# Autoimmune Diseases of the Eyelid Skin: Molecular Pathways, Clinical Manifestations, and Therapeutic Insights

**DOI:** 10.3390/ijms262311730

**Published:** 2025-12-04

**Authors:** Wojciech Luboń, Małgorzata Luboń, Monika Sarnat-Kucharczyk, Mariola Dorecka, Dorota Wyględowska-Promieńska

**Affiliations:** 1Department of Ophthalmology, Faculty of Medical Sciences, Medical University of Silesia, 40-514 Katowice, Polandmdorecka@sum.edu.pl (M.D.); dwygledowska@sum.edu.pl (D.W.-P.); 2Department of Ophthalmology, Professor K. Gibiński University Clinical Center, Medical University of Silesia, 40-514 Katowice, Poland; kozikowskamalg@gmail.com

**Keywords:** eyelid inflammation, autoimmune skin diseases, ocular cicatricial pemphigoid, microbiome, cytokine signaling, biologic therapy, targeted immunotherapy

## Abstract

The eyelid skin represents a unique anatomical and immunological interface between the external environment and the ocular surface. Due to its structural delicacy, dense vascularization, and continuous exposure to microbial and environmental antigens, it is a primary target of inflammatory and autoimmune processes. This review aims to synthesize current molecular insights into eyelid skin inflammation, with particular emphasis on autoimmune mechanisms. We discuss autoimmune diseases such as ocular cicatricial pemphigoid, pemphigus, discoid and systemic lupus erythematosus, and thyroid-associated orbitopathy, focusing on the roles of T helper cell subsets, pro-inflammatory cytokines (IL-1β, IL-6, IL-17, TNF-α), and autoantibody-mediated complement activation. We further address the contribution of the periocular microbiome and meibomian gland dysfunction. Diagnostic advances, including confocal microscopy, in vivo molecular imaging, and tear proteomics, are highlighted alongside emerging targeted therapies such as biologics and small molecules directed at IL-17, TNF-α, and B-cell activity. Finally, we propose future perspectives for precision medicine approaches, integrating omics technologies and microbiome-based therapies to advance personalized management of eyelid skin inflammation.

## 1. Introduction

The eyelid represents a unique anatomical and immunological structure that not only protects the ocular surface but also contributes to the homeostasis of the anterior segment of the eye. Unlike other cutaneous sites, eyelid skin is exceptionally thin, richly vascularized, and continuously exposed to mechanical stress from blinking as well as environmental insults such as ultraviolet radiation, allergens, microorganisms, and airborne pollutants. This distinctive microenvironment renders the eyelid particularly susceptible to localized dermatological disorders and systemic autoimmune diseases with periocular involvement [1].

Epidemiologically, eyelid dermatoses encompass a broad spectrum of inflammatory and allergic conditions, including atopic dermatitis, seborrheic dermatitis, and contact dermatitis, which are far more common in clinical practice than autoimmune disorders. Although autoimmune diseases of the eyelid are comparatively uncommon, they represent clinically significant entities characterized by immune dysregulation and potential systemic involvement [1,2]. The current review focuses exclusively on these autoimmune entities, highlighting their unique pathophysiology and clinical relevance.

Among non-autoimmune inflammatory conditions, periorbital (periocular) dermatitis represents one of the most frequent dermatoses affecting the eyelid region. It typically manifests as erythema, desquamation, and pruritus around the eyelids, often triggered by topical medications, cosmetics, or contact allergens. Chronic exposure may lead to lichenification and eyelid edema. Other common inflammatory diseases in this region include atopic and seborrheic dermatitis, which may coexist with ocular surface inflammation and mimic early autoimmune presentations [3,4].

The eyelid is among the most vascularized regions of the body, supplied by both the internal and external carotid arterial systems through the medial and lateral palpebral arteries. A dense superficial vascular plexus located within the thin dermis facilitates efficient thermoregulation but also predisposes to visible vascular changes. In inflammatory and autoimmune diseases, endothelial activation, cytokine-driven vasodilation, and chronic tissue remodeling can manifest clinically as telangiectasias of the eyelid margin and periocular skin. These vascular features are often early diagnostic indicators of chronic inflammation and immune dysregulation [1,2,3].

A wide range of inflammatory and autoimmune conditions affect the eyelid and its adnexal structures, which may present as early clinical manifestations or indicators of systemic disease. Notable examples include ocular cicatricial pemphigoid (OCP), pemphigus vulgaris, bullous pemphigoid, thyroid-associated orbitopathy (TAO), and lupus erythematosus. Beyond cosmetic disfigurement, chronic eyelid inflammation may lead to malpositions, trichiasis, and secondary ocular surface damage, ultimately resulting in vision-threatening complications [2,5].

On a molecular level, the immunopathogenesis of eyelid inflammation reflects complex interactions among epithelial barrier disruption, autoreactive lymphocytes, autoantibody-mediated complement activation, and cytokine-driven inflammatory cascades. Dysregulated Th1, Th2, and Th17 pathways, combined with impaired regulatory T-cell activity, underlie many autoimmune eyelid diseases. Elevated levels of pro-inflammatory mediators—including interleukin (IL)-1β, IL-6, IL-17, tumor necrosis factor alpha (TNF-α), and interferon gamma (IFN-γ)—have been identified in both tissue biopsies and tear fluid [2,6]. These cytokines orchestrate chronic tissue remodeling, fibrosis, and glandular dysfunction, including meibomian gland dropout and lacrimal gland infiltration, thereby exacerbating ocular surface instability [6].

The periocular microbiome adds another dimension to this pathophysiology. Commensal organisms such as *Demodex mites* and *Corynebacterium* species can trigger or perpetuate inflammation through antigen presentation and innate immune activation. The interplay between microbial dysbiosis and host immunity provides new insights into chronic eyelid conditions such as blepharitis, as well as their overlap with systemic autoimmunity [7].

Clinically, eyelid inflammation may precede or parallel systemic autoimmune disease. In OCP, autoantibodies targeting basement membrane zone components (e.g., BP180, laminin-332) drive conjunctival fibrosis and cicatricial eyelid changes. In TAO, orbital and periorbital fibroblast activation, adipogenesis, and inflammatory remodeling result in eyelid retraction and edema. Likewise, systemic lupus erythematosus and Sjögren’s syndrome frequently manifest periocular involvement ranging from cutaneous rashes to chronic meibomian gland dysfunction. Recognizing these early signs is crucial, as untreated periocular inflammation often heralds progressive ocular morbidity [8,9,10].

Although many studies have investigated autoimmune mechanisms in cutaneous and systemic diseases, direct evidence focusing on the eyelid and periocular region remains limited. Recent reports have described distinct local immune activation patterns and vascular remodeling specific to eyelid tissue, supporting its unique role as both a target and indicator of systemic immune dysregulation [11,12,13]. These studies highlight the importance of integrating eyelid-specific data into the broader understanding of autoimmune skin disease pathogenesis.

Therapeutic strategies for autoimmune eyelid inflammation are rapidly evolving. Traditional immunosuppressive approaches—including systemic corticosteroids and antimetabolites—remain standard but are limited by adverse effects. Advances in biologic agents, such as B-cell-depleting therapies (rituximab), TNF-α inhibitors, and monoclonal antibodies targeting IL-17 and IL-23, have significantly expanded the therapeutic landscape. In addition, innovative diagnostic modalities—including tear proteomics and molecular imaging—offer new opportunities for personalized medicine [14,15].

Taken together, the increasing recognition of eyelid involvement in autoimmune diseases underscores the need for a comprehensive review that integrates molecular mechanisms, clinical manifestations, and therapeutic advances. This article is a narrative literature review, synthesizing current molecular and clinical evidence across ophthalmology, dermatology, and immunology, with particular emphasis on autoimmune mechanisms underlying eyelid skin inflammation. By highlighting the eyelid as a model structure for studying the skin–immune–ocular interface, such an approach may facilitate earlier diagnosis, guide therapeutic innovation, and ultimately prevent irreversible ocular morbidity.

## 2. Molecular Pathways of Eyelid Skin Inflammation

Eyelid inflammation arises from a complex interplay of immune and non-immune pathways. Central processes include epithelial barrier disruption, activation of adaptive immunity, and cytokine-driven signaling cascades. Collectively, these mechanisms promote chronic tissue remodeling, fibrosis, and glandular dysfunction, which characterize autoimmune-mediated diseases of the eyelid skin and ocular adnexa.

### 2.1. Cytokine and T-Cell Pathways

Dysregulated T helper (Th) subsets dominate the immune landscape of eyelid inflammation. Increased activity of Th1 and Th17 cells has been demonstrated in both OCP and Sjögren’s syndrome, accompanied by elevated levels of IL-1β, IL-6, IL-17, TNF-α, and IFN-γ in periocular tissues. IL-17 is a key driver of fibroblast activation and conjunctival scarring, whereas TNF-α and IFN-γ sustain chronic inflammatory loops [2,8]. In contrast, impaired regulatory T-cell function reduces immune tolerance and permits uncontrolled autoimmunity [5].

### 2.2. Autoantibody-Mediated Complement Activation

Humoral immunity plays a central role in several autoimmune eyelid diseases. In OCP, autoantibodies targeting basement membrane zone components such as BP180, BP230, and laminin-332 initiate complement activation and neutrophil recruitment, resulting in subepithelial inflammation and fibrosis [7]. In pemphigus vulgaris and bullous pemphigoid, IgG autoantibodies against desmogleins and hemidesmosomal proteins lead to acantholysis and blistering, which may also involve the eyelid skin.

### 2.3. Fibroblast and Adipocyte Signaling in Thyroid Eye Disease

In TAO, orbital and periocular fibroblasts aberrantly express thyroid-stimulating hormone receptor (TSHR) and insulin-like growth factor-1 receptor (IGF-1R). Activation of these receptors drives excessive glycosaminoglycan production, adipogenesis, and tissue edema, which clinically manifest as eyelid retraction and periorbital swelling [6]. Genetic studies implicate polymorphisms in co-stimulatory genes, further highlighting the role of immune–stromal interactions in TAO pathogenesis [14].

### 2.4. Role of Microbiome and Innate Immunity

Increasing evidence supports a contribution of the periocular microbiome to eyelid inflammation. Overgrowth of *Demodex mites* and altered colonization by *Corynebacterium* species can activate toll-like receptor (TLR) pathways, inducing IL-8 and matrix metalloproteinases that damage the eyelid margin [16]. Such dysbiosis-driven inflammation provides a mechanistic link between chronic blepharitis and systemic autoimmunity, where innate immune activation lowers the threshold for adaptive responses [1,16].

Recent metagenomic and culture-based studies have broadened the understanding of the eyelid microbiome, revealing a diverse community that includes *Staphylococcus epidermidis*, *Staphylococcus aureus*, *Corynebacterium*, and *Propionibacterium species*, alongside fungal genera such as *Malassezia* and *Aspergillus*. These studies were observational and sequencing-based rather than interventional clinical trials; therefore, ClinicalTrials.gov identifiers are not applicable. Imbalances within these microbial populations can promote chronic inflammation through the activation of innate immune receptors and altered lipid metabolism at the eyelid margin. Ongoing investigations are exploring how these bacterial and fungal communities interact with tear film components, meibomian gland function, and local immune responses, particularly in the context of autoimmune blepharitis and ocular cicatricial pemphigoid. These findings highlight that eyelid microbiome dysregulation is not merely secondary to inflammation but may actively contribute to disease initiation and chronicity. In healthy eyelid skin, a balanced commensal flora dominated by *Staphylococcus epidermidis* and *Corynebacterium* species supports immune tolerance and lipid homeostasis. In periocular dermatitis and chronic blepharitis, this equilibrium is disrupted, leading to pathogenic overgrowth and sustained local inflammation [17,18].

### 2.5. Oxidative Stress and Epithelial Barrier Dysfunction

Oxidative stress acts as a potent amplifier of inflammatory signaling. In both TAO and OCP, reactive oxygen species (ROS) upregulate NF-κB activity, enhancing transcription of pro-inflammatory cytokines and adhesion molecules. This cascade compromises epithelial barrier integrity, perpetuates inflammation, and contributes to irreversible tissue remodeling [15,19].

### 2.6. Summary

Collectively, these molecular insights underscore that eyelid inflammation is not an isolated process but a reflection of systemic immune dysregulation. They also point to promising therapeutic targets, including IL-17, TNF-α, T-cell co-stimulatory pathways, and microbiome-directed interventions, which may shape the next generation of precision therapies.

While many immunopathogenic mechanisms described in this section parallel those observed in systemic autoimmune diseases, the direct evidence specific to eyelid tissues remains limited. Much of the current understanding derives from extrapolation of data obtained from cutaneous, mucosal, or conjunctival models. Although such analogies provide valuable mechanistic context, further dedicated studies focusing on eyelid-specific immune regulation, vascular responses, and stromal remodeling are needed to confirm whether these systemic paradigms fully apply to periocular pathology.

Despite considerable overlap among immune and molecular mechanisms across autoimmune diseases, direct evidence specific to the eyelid skin and tear film remains limited. Most mechanistic data have been derived from studies of cutaneous, conjunctival, or lacrimal tissues, and their extrapolation to the eyelid should be interpreted with caution. Processes such as complement activation and T-cell-mediated fibrosis have been directly demonstrated in conditions like ocular cicatricial pemphigoid and lupus erythematosus, whereas other pathways—such as IL-17-driven inflammation and oxidative stress—are primarily supported by indirect evidence. Further studies focused on eyelid- and tear-fluid-based immunopathology are needed to confirm these mechanistic parallels.

## 3. Autoimmune Diseases Involving Eyelid Skin and Ocular Adnexa

Autoimmune diseases represent a major cause of eyelid skin inflammation. Their manifestations in the ocular adnexa extend beyond cosmetic changes, frequently leading to significant visual morbidity. Pathogenesis involves diverse immune mechanisms, including autoantibody-mediated complement activation, T-cell-driven cytokine responses, and stromal tissue remodeling. Below, we summarize the principal autoimmune diseases with eyelid involvement, emphasizing their molecular basis, clinical presentation, and therapeutic implications [11].

While many studies describing these autoimmune disorders are based on systemic or generalized cutaneous frameworks, several reports have documented specific periocular and eyelid manifestations. Recent case series and clinicopathologic analyses confirm that autoimmune mechanisms may directly target the eyelid skin, conjunctiva, and adnexal structures, resulting in characteristic inflammatory, fibrotic, or vascular changes distinct from those observed in non-ocular tissues [3,11,13].

### 3.1. Ocular Cicatricial Pemphigoid (OCP)

OCP is a chronic, progressive autoimmune blistering disease characterized by subepithelial fibrosis primarily affecting the conjunctival mucosa, but it may also extend to the eyelid skin and other mucous membranes. Autoantibodies against basement membrane zone antigens (BP180, BP230, laminin-332) activate complement and recruit neutrophils, resulting in inflammation and fibrosis [7]. Clinically, this process causes conjunctival scarring, symblepharon formation, and cicatricial entropion with trichiasis, severely compromising the ocular surface. Eyelid skin involvement may include erythema, scarring, and keratinization. At the molecular level, Th2 and Th17 cytokines, particularly IL-17, drive fibroblast activation and extracellular matrix deposition. Without timely immunosuppression, OCP progresses relentlessly, highlighting the need for early recognition. In OCP, chronic conjunctival and eyelid inflammation may be accompanied by superficial telangiectatic changes due to persistent vascular remodeling [2,20].

### 3.2. Pemphigus Vulgaris and Bullous Pemphigoid

Pemphigus vulgaris is mediated by IgG autoantibodies targeting desmoglein-1 and desmoglein-3, leading to acantholysis and intraepithelial blistering. Although the oral mucosa is most frequently involved, periocular skin may also exhibit erosions and ulcerations. By contrast, bullous pemphigoid involves hemidesmosomal proteins, producing subepidermal blistering. Eyelid manifestations include erythematous plaques, crusting, and secondary infections, which can mimic chronic blepharitis and delay diagnosis [21,22]. Direct immunofluorescence remains the established diagnostic method for confirming autoimmune blistering diseases.

### 3.3. Thyroid-Associated Orbitopathy (TAO)

TAO, or thyroid eye disease, is the most common autoimmune disorder affecting the eyelids. Autoreactive T cells target TSHR and IGF-1R on orbital fibroblasts, driving glycosaminoglycan production, adipogenesis, and edema [6]. Clinically, eyelid retraction occurs in up to 90% of cases [8], accompanied by periorbital edema and dermopathy. Oxidative stress and NF-κB activation amplify cytokine release, perpetuating tissue remodeling. Recent trials demonstrate that teprotumumab, an IGF-1R inhibitor, effectively reduces eyelid and orbital inflammation [15,23], underscoring the translational impact of molecular discoveries.

### 3.4. Systemic Lupus Erythematosus (SLE)

SLE is a systemic autoimmune disease with frequent cutaneous manifestations, including in the periocular region. Discoid lesions and malar rashes may extend to the eyelids, presenting with erythema, scaling, and dyspigmentation. Eyelid involvement in discoid lupus erythematosus (DLE) and systemic lupus erythematosus (SLE) has been increasingly recognized in both clinical and histopathological studies [11,12,13]. In DLE, chronic interface dermatitis and localized inflammation produce erythematous, scaly, and atrophic plaques along the eyelid margins, sometimes leading to lash loss or cicatricial changes that can mimic chronic blepharitis [24]. In SLE, vascular instability and immune complex deposition contribute to characteristic periocular telangiectasias and erythematous lesions, reflecting immune-mediated vascular injury. Histologically, immune complex deposition at the dermoepidermal junction activates complement and recruits inflammatory infiltrates, while Type I interferon pathways, particularly IFN-α-inducible genes, play a central role in perpetuating cutaneous inflammation [13,19,25]. Recognizing these features is clinically important, as eyelid lesions may precede or parallel systemic disease activity and contribute to ocular surface instability.

### 3.5. Sjögren’s Syndrome

Sjögren’s syndrome primarily affects the exocrine glands. Periocular manifestations result from lymphocytic infiltration of the lacrimal glands, leading to aqueous tear deficiency, and of the meibomian glands, causing lipid layer instability. Chronic eyelid margin inflammation drives meibomian gland dysfunction, blepharitis, and keratoconjunctivitis sicca. Molecular drivers include Th17 cytokines (IL-17, IL-22) and B-cell hyperactivity. Pediatric cases, though rare, often present with ocular signs that precede systemic disease [8,26].

### 3.6. Sarcoidosis and Granulomatous Disorders

Sarcoidosis is a systemic granulomatous disease in which periocular involvement occurs in 10–15% of patients, manifesting as eyelid nodules, erythema, or diffuse edema. Histopathology demonstrates non-caseating granulomas composed of epithelioid histiocytes and multinucleated giant cells. Cytokines such as TNF-α, IL-2, and IFN-γ drive granuloma formation, linking sarcoidosis to broader Th1-mediated immune pathways [12,21]. Eyelid sarcoid lesions may clinically resemble malignancies, complicating differential diagnosis.

### 3.7. Other Rare Autoimmune and Autoinflammatory Syndromes

Several rare disorders can involve the eyelids. Vacuoles, E1 enzyme, X-linked, autoinflammatory, and somatic mutation (VEXAS) syndrome, caused by somatic mutations in *UBA1*, presents with systemic autoinflammation and ocular adnexal manifestations, including eyelid swelling and orbital inflammation Drug-induced eyelid dermatitis, often triggered by systemic or topical medications, represents a type IV (T-cell-mediated) hypersensitivity reaction involving the periocular skin, rather than a classical autoimmune or autoinflammatory disorder. Such reactions have been comprehensively characterized in the recent European Academy of Allergy and Clinical Immunology (EAACI) Task Force position paper on drug-induced hypersensitivity [27].

Immune-related adverse events induced by immune checkpoint inhibitors (e.g., anti-PD-1, anti-PD-L1, anti-CTLA-4 therapies) may also lead to periocular inflammation or granulomatous lesions. These reactions encompass a broad immunologic spectrum—from allergic or inflammatory responses to autoimmune-like phenomena, in which therapeutic checkpoint blockade activates autoreactive lymphocytes and tissue-specific inflammation mimicking autoimmune disease [5,10,22].

### 3.8. Dermatomyositis

Dermatomyositis is an autoimmune connective tissue disease characterized by chronic inflammation of the skin and muscles. Its cutaneous manifestations frequently involve the periocular region, where the classic heliotrope rash—a violaceous discoloration with periorbital edema—is a hallmark diagnostic feature. This periocular involvement may precede or accompany systemic symptoms, including proximal muscle weakness. On a molecular level, dermatomyositis is associated with immune complex deposition, complement-mediated microangiopathy, and the presence of myositis-specific autoantibodies such as anti-Mi-2 and anti-MDA5. Recognition of eyelid and periorbital changes is crucial, as they may serve as early indicators of systemic disease and guide timely immunosuppressive intervention [3,28].

### 3.9. Summary

Autoimmune diseases involving the eyelid skin encompass a diverse group of disorders unified by chronic immune dysregulation and overlapping molecular pathways. Recognition of periocular manifestations is critical, as they may represent the first sign of systemic disease. Telangiectatic changes of the eyelid and periocular skin, resulting from chronic vascular remodeling, may serve as visible indicators of underlying autoimmune inflammation and should be considered relevant clinical findings across several disease entities. Advances in molecular diagnostics and targeted therapies—including B-cell depletion, cytokine inhibition, and receptor blockade—hold promise for earlier intervention and improved patient outcomes. The diseases summarized in Table 1 encompass autoimmune and immune-mediated disorders that may involve the eyelid skin and periocular region. The selection reflects both well-established and less common conditions, illustrating the diversity of immune mechanisms contributing to periocular inflammation. This framework aims to provide a comprehensive overview of clinically and mechanistically relevant entities affecting the eyelid and its adnexal structures. It is important to distinguish between eyelid skin lesions and conjunctival mucosal changes, as they may reflect different disease mechanisms and require distinct diagnostic and therapeutic approaches.

## 4. Clinical Implications and Diagnostics

Accurate diagnosis of autoimmune-mediated eyelid inflammation requires a multimodal strategy that combines clinical evaluation with histopathology, immunological assays, and advanced imaging. In clinical evaluation, eyelid skin lesions—such as erythema, scarring, and telangiectasia—should be differentiated from mucosal involvement of the conjunctiva, which presents with fibrosis, symblepharon, or forniceal shortening. Because eyelid manifestations may precede systemic disease, early recognition is essential to prevent irreversible ocular morbidity [6,13].

### 4.1. Clinical Examination and Differential Diagnosis

Slit-lamp biomicroscopy remains the cornerstone of periocular assessment. Eyelid margin changes—such as telangiectasia, scarring, trichiasis, and keratinization—are highly suggestive of autoimmune cicatricial disease, including OCP. Differentiation from chronic blepharitis or infectious etiologies is critical, as delayed recognition permits progression of fibrosis [2,7].

Malignant eyelid neoplasms, including basal cell carcinoma, squamous cell carcinoma, and sebaceous gland carcinoma, should also be considered in the differential diagnosis, as they can clinically resemble chronic autoimmune or inflammatory conditions of the eyelid margin. Careful inspection and biopsy of suspicious lesions are essential to exclude neoplastic processes before initiating long-term immunosuppressive therapy [29].

Standardized clinical grading systems, such as the Foster staging system for OCP and the Clinical Activity Score (CAS) for TAO, provide objective measures of severity and guide treatment [2,9].

### 4.2. Histopathology and Direct Immunofluorescence

Biopsy of conjunctiva or eyelid skin remains the established reference method for definitive diagnosis in autoimmune blistering diseases, as endorsed by current dermatologic and ophthalmologic consensus statements [30,31]. In OCP, subepithelial fibrosis with lymphocytic infiltration is typical, while direct immunofluorescence reveals linear deposition of IgG, IgA, and complement (C3) along the basement membrane zone. In pemphigus vulgaris, intraepidermal acantholysis is evident, with intercellular IgG deposition against desmogleins. Immunopathological confirmation is therefore critical for distinguishing autoimmune disease from infectious or neoplastic mimics [26,32].

### 4.3. Eyelid Biopsy: Indications, Risks, and Contraindications

Eyelid biopsy is an important diagnostic procedure in suspected autoimmune or inflammatory diseases affecting periocular tissues. It is indicated when the clinical presentation is atypical, when differentiation from infection, neoplasia, or other chronic dermatoses is required, or when histopathologic confirmation is necessary before initiating systemic immunosuppressive therapy. When diagnostic lesions exist at other cutaneous sites, biopsy from these areas is generally preferred to reduce the risk of eyelid scarring or functional impairment [29,32].

Biopsy of the eyelid should be avoided in cases of active infection, severe inflammation, or marked edema, as these conditions increase the risk of poor wound healing and secondary fibrosis. Caution is advised in patients with diabetes, vascular insufficiency, or chronic corticosteroid use, in whom epithelial closure may be delayed. If eyelid biopsy is required, a small, carefully planned incision from the lateral or tarsal area minimizes cosmetic and functional morbidity while providing adequate diagnostic tissue. When performed under proper aseptic conditions, healing is usually uncomplicated, and histopathologic evaluation remains the established diagnostic reference for confirming autoimmune blistering and cicatricial diseases [29,33].

### 4.4. Advanced Imaging Modalities

Innovations in ophthalmic imaging have expanded diagnostic capabilities. In vivo confocal microscopy (IVCM) enables high-resolution visualization of epithelial alterations and inflammatory infiltrates, providing a non-invasive complement to biopsy. Anterior segment optical coherence tomography (AS-OCT) offers cross-sectional views of eyelid margins and conjunctival scarring, allowing longitudinal monitoring in OCP and TAO [6,34]. These modalities are increasingly integrated into clinical trials as quantitative endpoints for disease progression and therapeutic efficacy.

### 4.5. Molecular Biomarkers in Tears and Serum

Molecular profiling of tears has revealed elevated levels of IL-6, IL-17, TNF-α, and matrix metalloproteinase-9 (MMP-9) in patients with OCP, TAO, and Sjögren’s syndrome, correlating with disease severity. Serum biomarkers—such as anti-BP180 and anti-laminin-332 autoantibodies in OCP and thyroid-stimulating immunoglobulins (TSI) in TAO—are increasingly used for diagnosis and monitoring. Genetic polymorphisms in co-stimulatory pathways and markers of oxidative stress have been linked to more aggressive phenotypes. These molecular signatures may help stratify patients and tailor therapeutic choices [35,36,37,38].

Proteomic analyses of tear fluid have identified disease-specific molecular signatures in autoimmune eyelid disorders. Altered levels of proteins such as lactoferrin, lipocalin-1, and MMP-9 have been reported in ocular cicatricial pemphigoid, thyroid-associated orbitopathy, and Sjögren’s syndrome, correlating with inflammation severity. However, tear proteomics remains non-standardized due to differences in sample handling and analytical protocols, which limit reproducibility. Ongoing studies aim to establish unified methodologies to enable proteomic biomarkers to serve as reliable diagnostic and monitoring tools in eyelid inflammation [35,37].

### 4.6. Integration of Diagnostics into Personalized Medicine

The integration of histopathology, imaging, and molecular biomarkers is reshaping diagnostic paradigms. Artificial intelligence (AI)-based image analysis holds promise for improving diagnostic accuracy by enabling the stratification of patients toward precision therapies. Recent AI-based studies in ophthalmology have demonstrated accurate automated detection of eyelid margin abnormalities, meibomian gland dropout, and periocular inflammation using deep-learning algorithms applied to slit-lamp and meibography images. These approaches offer objective, reproducible assessments that can complement clinical evaluation and support earlier recognition of eyelid involvement in autoimmune diseases [39,40]. Furthermore, multi-omics approaches—including genomics, proteomics, and metabolomics—may enable early identification of high-risk patients and support biomarker-driven treatment algorithms [41,42].

Validated questionnaires such as the Ocular Surface Disease Index (OSDI) and the Graves’ Orbitopathy Quality of Life (GO-QOL) scale can aid in assessing patient-reported symptoms and quality of life in autoimmune eyelid disease. Their use helps standardize outcome evaluation and supports personalized management [43].

### 4.7. Summary

The diagnostic approach to autoimmune-mediated eyelid inflammation increasingly relies on a multimodal framework. While biopsy and immunofluorescence remain central, advances in imaging and biomarker discovery are enhancing diagnostic precision and disease monitoring. Together, these innovations pave the way toward personalized medicine strategies, offering the potential to transform clinical outcomes in patients with autoimmune eyelid disease (Table 2).

## 5. Therapeutic Strategies

Management of autoimmune-mediated eyelid inflammation requires an individualized approach that addresses both local manifestations and systemic disease activity.

In mild or localized cases, particularly when inflammation is confined to the eyelid margin, topical therapy may be sufficient. Topical corticosteroid or calcineurin inhibitor ointments, combined with eyelid hygiene and anti-inflammatory lubricants, can effectively control symptoms without the need for systemic immunosuppression. Eyelid hygiene typically includes the gentle cleansing of the eyelid margins with sterile wipes or diluted non-irritant solutions to remove crusts, debris, and microbial biofilm. Regular warm compresses may be applied to soften meibomian gland secretions, reduce obstruction, and restore the stability of the tear film microenvironment [44]. Eyelid hygiene recommendations are summarized in Box 1.

Box 1Eyelid Hygiene Recommendations.
Cleanse eyelid margins twice daily using sterile wipes or diluted, non-irritant cleansing solutions.Apply warm compresses for 5–10 min to soften meibomian gland secretions and improve lipid flow.Gently massage the eyelid margin to promote gland clearance.Avoid harsh soaps, alcohol-based solutions, or cosmetic products near the eyelid margin.Maintain consistent daily hygiene even during asymptomatic periods to prevent recurrence.


When topical corticosteroids are used, low- to medium-potency agents such as hydrocortisone 1% or fluorometholone 0.1% are preferred due to the thinness and high absorption capacity of eyelid skin. Treatment courses should remain short—generally limited to one to two weeks—to minimize adverse effects, including skin atrophy, telangiectasia, and increased intraocular pressure. In cases requiring prolonged control, transition to steroid-sparing agents such as tacrolimus or pimecrolimus is recommended to maintain efficacy while reducing local toxicity. Systemic treatment is indicated only when periocular inflammation reflects active or progressive systemic autoimmune disease [13,19,45].

Advances in molecular insights have reshaped therapeutic paradigms, particularly in OCP, pemphigus, and TAO, where periocular involvement is prominent.

In alignment with current international standards, therapeutic recommendations for eyelid inflammation and associated autoimmune disorders should adhere to evidence-based consensus frameworks. The TFOS DEWS III Report provides an updated, stepwise protocol for the management of blepharitis and meibomian gland dysfunction, emphasizing eyelid hygiene, anti-inflammatory therapy, and adjunctive procedures such as thermal pulsation and light-based modalities in refractory cases [44].

Similarly, the European Dermatology Forum (EDF) and the European Academy of Dermatology and Venereology (EADV) guidelines for autoimmune blistering diseases outline early initiation of immunosuppressive therapy, integration of steroid-sparing agents, and the introduction of biologic or small-molecule therapies—such as rituximab and JAK inhibitors—in resistant disease [30].

For periocular manifestations of thyroid-associated orbitopathy, the European Group on Graves’ Orbitopathy (EUGOGO) guidelines recommend disease activity-based management, selenium supplementation, smoking cessation, and targeted inhibition of the IGF-1R using teprotumumab for active, moderate-to-severe disease [31].

Incorporating these frameworks ensures that the therapeutic section of this review remains aligned with internationally accepted, evidence-based standards for the management of blepharitis, autoimmune blistering disease, and thyroid-associated orbitopathy.

Although these treatment recommendations are derived from robust systemic and dermatologic data, direct evidence regarding their efficacy in eyelid-specific inflammation remains limited. Most published studies describe improvements in periocular symptoms as secondary observations rather than primary endpoints. Nevertheless, isolated clinical reports support the beneficial effects of immunosuppressive and biologic therapies—such as corticosteroids, rituximab, and JAK inhibitors—on eyelid inflammation in autoimmune conditions, suggesting translational applicability of systemic therapeutic principles to periocular disease [17,18,30].

### 5.1. Conventional Immunosuppressive Therapy

Systemic corticosteroids remain the first-line treatment for acute exacerbations, rapidly suppressing inflammation and preventing fibrotic progression. However, their long-term use is constrained by adverse effects such as skin atrophy, infection, osteoporosis and ocular hypertension. For chronic control, antimetabolites—including azathioprine, mycophenolate mofetil, and methotrexate—serve as steroid-sparing agents in OCP and pemphigus. Cyclophosphamide is reserved for severe, vision-threatening cases with progressive scarring [46,47,48].

Topical therapies complement systemic treatment. Corticosteroid ointments and calcineurin inhibitors (tacrolimus, cyclosporine A) provide local control of eyelid inflammation [18,46]. Because of the thinness and high absorption capacity of eyelid skin, topical corticosteroids should be used only short term and with caution. Prolonged use may lead to skin atrophy, telangiectasia, and elevated intraocular pressure, as reported in studies on eyelid dermatitis and periocular corticosteroid exposure [27,44,45]. Topical tacrolimus may help reduce erythema and scarring in OCP and chronic periocular dermatitis, showing a favorable safety profile limited mainly to transient burning or stinging sensations after application [18,47]. Cyclosporine eye drops improve eyelid margin inflammation and tear film stability in autoimmune blepharitis and Sjögren’s syndrome, supported by clinical evidence demonstrating reduced ocular surface inflammation and improved tear film parameters [8,26,47]. Combining systemic and local therapy enhances efficacy while limiting systemic toxicity.

### 5.2. Biologic Agents

Biologic therapies have revolutionized management of refractory autoimmune disease. Although no biologic agent is specifically licensed for disorders limited to the eyelid, several have been approved for systemic autoimmune conditions with frequent periocular involvement.

Rituximab, an anti-CD20 monoclonal antibody, is approved for pemphigus vulgaris and has demonstrated efficacy in OCP by depleting autoreactive B cells [15,29,49]. TNF-α inhibitors such as infliximab and adalimumab are occasionally used in pemphigoid and lupus with periocular inflammation, though clinical responses remain variable [30,48].

Teprotumumab, an IGF-1R-blocking antibody, is the first targeted biologic therapy approved for TAO, demonstrating significant improvement in eyelid retraction, proptosis, and orbital inflammation, as shown in clinical studies [23,50]. Building on these outcomes, current therapeutic strategies increasingly incorporate biologic agents as part of individualized management approaches for periocular autoimmune disease [6,9,49]. TAO has therefore emerged as a key indication for biologic therapy, highlighting the translational relevance of systemic biologics for periocular autoimmune disease.

Emerging biologics targeting interleukin pathways, including IL-6 (tocilizumab) and IL-17 inhibitors, are under active investigation and may further expand precision therapy options for refractory cases [6,9,49].

### 5.3. Small Molecules and Targeted Oral Therapies

Janus kinase (JAK) inhibitors such as baricitinib and tofacitinib block intracellular cytokine signaling (IL-6, IL-17, IFN-γ), reducing inflammation and stabilizing scarring in refractory autoimmune blistering diseases. Their use has been reported in case studies and early clinical observations demonstrating symptomatic improvement [51,52,53]. Their oral administration and multi-cytokine blockade make them attractive for long-term therapy, though monitoring is required for infection and thrombotic risk [51].

Other small molecules, such as apremilast (PDE-4 inhibitor), have shown efficacy in cutaneous autoimmune disease and may benefit periocular lesions. Current evidence is largely limited to case reports and small clinical observations yet suggests potential roles in chronic, treatment-resistant inflammation [54].

Several targeted small-molecule therapies have recently gained attention in the management of autoimmune and inflammatory diseases involving the skin and ocular adnexa. Among them, JAK inhibitors such as tofacitinib and baricitinib, and phosphodiesterase-4 inhibitors such as apremilast, have demonstrated promising immunomodulatory and anti-inflammatory effects in mucocutaneous conditions. These therapeutic advances may help refine safety profiles, optimize dosing, and support the development of more selective, eyelid-focused treatment approaches [47,51,52,53,54].

### 5.4. Local Adjunctive Therapies

Local interventions are crucial for structural complications. Trichiasis from cicatricial entropion may require epilation, electrolysis, or cryotherapy, while severe cases necessitate reconstructive surgery, including mucous membrane or amniotic membrane grafts (AMG) [2].

Device-based adjuncts, such as thermal pulsation and meibomian gland expression, improve lipid secretion and reduce chronic eyelid inflammation. Light-based modalities—including intense pulsed light (IPL) and low-level light therapy (LLLT)—have shown benefit in chronic blepharitis overlapping with autoimmune disease, reducing vascular congestion and inflammatory cytokine activity. These procedures are generally well tolerated, with transient erythema, mild discomfort, or temporary ocular surface irritation being the most commonly reported adverse effects [55,56].

### 5.5. Lifestyle and Supportive Interventions

Supportive measures are integral to long-term care. Selenium supplementation improves outcomes in TAO, while antioxidant-rich diets (e.g., Mediterranean diet) reduce systemic oxidative stress and NF-κB activation. Smoking cessation is essential, as smoking is a major risk factor for TAO progression and poor therapeutic response [15,16].

Adjunctive ocular surface care (warm compresses, eyelid hygiene, preservative-free lubricants) improves comfort and may reduce secondary infections in autoimmune blepharitis by restoring meibomian gland function and limiting microbial overgrowth at the eyelid margin [16,17].

### 5.6. Toward Personalized Immunotherapy

Molecular diagnostics are increasingly supporting biomarker-driven therapeutic choices. Tear proteomic studies identify IL-6, IL-17, TNF-α, and MMP-9 as correlates of severity, while genetic profiling (e.g., co-stimulatory polymorphisms) may predict aggressive phenotypes [14]. Patients with Th17-dominant disease may respond best to IL-17 inhibition, whereas B-cell-driven disease benefits from rituximab. Microbiome-targeted approaches, including probiotics or selective antibiotics, represent emerging strategies for periocular autoimmune control [14,56].

### 5.7. Summary

Current management of autoimmune-mediated eyelid inflammation spans conventional immunosuppression, biologics, small molecules, and adjunctive local therapies. The shift from broad immunosuppression toward biomarker-guided, precision interventions marks a paradigm change in treatment. This transition can be considered to be in its early developmental phase, with growing implementation of targeted biologics and molecular profiling gradually shaping more individualized treatment approaches. Integration of systemic, local, and lifestyle approaches—anchored in molecular profiling—promises improved efficacy, reduced toxicity, and a future of personalized immunotherapy for eyelid autoimmune disease.

While the therapeutic options for autoimmune eyelid inflammation continue to expand, the level of evidence supporting these interventions varies considerably. Most data on eyelid-specific outcomes derive from small case series or observational studies rather than randomized controlled trials [15,47]. The efficacy of systemic corticosteroids and antimetabolites remains supported mainly by retrospective analyses, while biologic therapies such as rituximab and teprotumumab are underpinned by moderate-quality evidence from targeted clinical studies. Reports on JAK inhibitors and IL-17/IL-23 antagonists are limited to isolated case descriptions and early-phase evaluations, warranting cautious interpretation [51,52,53]. Therefore, current management remains largely guided by extrapolation from broader autoimmune and dermatologic disease models rather than eyelid-specific trials, underscoring the need for prospective, controlled studies. Although many therapeutic principles are shared among autoimmune diseases, most data supporting these strategies originate from systemic or dermatologic disease contexts. The direct evidence for their efficacy in eyelid-specific inflammation is still emerging, and future research should validate whether these systemic paradigms fully translate to periocular disease.

## 6. Future Directions

The future management of autoimmune eyelid disease is expected to be shaped by precision medicine, integrating molecular diagnostics, targeted therapies, microbiome modulation, and digital health. While current approaches rely on broad immunosuppression and biologics, several research avenues —particularly in OCP and TAO—are driving the development of targeted and biomarker-based treatments [2,6,19].

### 6.1. Biomarker Development and Multi-Omics Approaches

Advances in high-throughput sequencing and proteomics are reshaping the understanding of eyelid inflammation. Tear proteomics has already identified cytokines and proteases—including IL-17, IL-6, TNF-α, and MMP-9—that correlate with disease severity in OCP, TAO, and Sjögren’s syndrome [56,57,58]. Future validation of these markers could enable earlier detection of aggressive phenotypes. Integrative multi-omics strategies combining genomics, transcriptomics, proteomics, and metabolomics are likely to uncover new pathways driving fibrosis and keratinization in autoimmune blistering diseases [57,58]. Integration of omics datasets with imaging may support risk stratification and prediction of therapeutic response.

### 6.2. Microbiome and Immune Modulation

The periocular microbiome represents an emerging therapeutic frontier. Dysbiosis, including overgrowth of *Demodex mites* and *Corynebacterium* species, has been implicated in chronic eyelid inflammation [1,16]. Future interventions may include selective antibiotics, probiotics, or microbiota transplantation to restore immune tolerance and reduce flares. Identification of microbial metabolites influencing T-cell differentiation could further illuminate the skin–immune interface at the eyelid [17,18].

### 6.3. Novel Targeted and Localized Therapies

Next-generation therapies are moving beyond current biologics. JAK inhibitors and IL-17/IL-23 antagonists are being explored for autoimmune blistering diseases, with topical formulations having been proposed for periocular application. Nanotechnology-based delivery platforms may enable localized immunomodulation, such as nanoparticle-encapsulated corticosteroids or siRNA against pro-fibrotic genes, minimizing systemic toxicity [53,59].

Recent publications describe the development of topical and nanoparticle-based formulations of JAK inhibitors, siRNA molecules, and other immunomodulatory agents for site-specific autoimmune inflammation. These emerging concepts highlight a broader translational shift toward precision, localized ocular and periocular therapies [60].

### 6.4. Artificial Intelligence and Digital Health

Artificial intelligence (AI) and digital technologies are poised to enhance diagnosis and monitoring. Machine learning applied to eyelid photography and anterior segment OCT can facilitate automated detection of early cicatricial changes. Recent AI applications in oculoplastic imaging and periocular analysis have demonstrated the feasibility of detecting eyelid malpositions, periocular edema, and inflammatory signs, suggesting potential use in autoimmune conditions [41]. Remote monitoring through smartphone-based imaging could extend disease tracking beyond clinic visits. AI tools also address unmet clinical needs by enabling objective, reproducible assessment of disease activity and supporting earlier intervention in chronic inflammatory disorders. Ultimately, AI-driven integration of imaging, clinical, and biomarker data may enable personalized treatment algorithms and prediction of disease flares [59,61].

### 6.5. Patient-Centered and Preventive Strategies

Future research must also prioritize patient-reported outcomes and prevention. Chronic eyelid inflammation contributes to functional impairment, cosmetic disfigurement, and psychological burden, reflecting significant unmet needs in quality of life and daily functioning among affected patients [36,62]. These impacts can be measured through validated ocular surface-related questionnaires and vision-specific quality-of-life indices. Addressing these dimensions is essential, as integrating patient-centered metrics into therapeutic decision-making may guide earlier intervention and improve long-term adherence to treatment. Clinical trials incorporating both molecular and quality-of-life endpoints will be essential. Moreover, genetic and biomarker-based screening could identify high-risk individuals, enabling early intervention before irreversible fibrosis occurs [46,62].

### 6.6. Summary

The field is moving decisively toward biomarker-guided, microbiome-informed, and digitally supported precision medicine. By integrating molecular research, targeted therapies, and patient-centered strategies, future approaches hold the potential to revolutionize outcomes and quality of life for individuals with autoimmune-mediated eyelid skin inflammation.

## 7. Conclusions

Autoimmune diseases of the eyelid skin represent a unique bridge between dermatology, ophthalmology, and immunology. Once considered secondary to systemic manifestations, periocular involvement is now recognized as a clinically and biologically significant feature. The delicate structure of the eyelid, coupled with constant environmental exposure, makes it a sensitive indicator of immune dysregulation. Recognizing eyelid inflammation as more than a local phenomenon is critical for improving diagnosis, therapy, and patient outcomes.

At the molecular level, cytokine imbalance, autoantibody-driven complement activation, and fibroblast remodeling underlie chronic eyelid inflammation. Dysregulated Th1, Th2, and Th17 pathways, together with impaired regulatory T-cell activity and microbial dysbiosis, explain the heterogeneity of clinical presentations. These findings emphasize the systemic dimension of eyelid disease.

Clinically, periocular inflammation is not only disfiguring but also often vision-threatening, with OCP, pemphigus, TAO, and Sjögren’s syndrome frequently beginning with subtle eyelid signs [2,5]. If unrecognized, these changes may progress to fibrosis, trichiasis, and severe ocular surface disease. The eyelid thus acts as a sentinel structure, signaling systemic activity and demanding multidisciplinary evaluation.

Therapeutically, the shift from broad immunosuppression to targeted approaches—exemplified by rituximab, teprotumumab, and JAK inhibitors—represents a major advance. Adjunctive surgical and device-based interventions expand treatment options, although challenges remain in ensuring accessibility, long-term safety, and personalization.

Looking ahead, biomarker-guided therapy, multi-omics integration, and AI-driven diagnostics promise earlier detection, tailored treatment, and prevention of irreversible damage. Incorporating patient-centered outcomes will be equally important, reflecting the profound psychosocial and visual impact of periocular disease.

In conclusion, autoimmune eyelid inflammation exemplifies the convergence of local and systemic immune dysregulation. Integration of molecular discoveries, advanced diagnostics, and targeted therapeutics heralds a new era of precision medicine in periocular autoimmune disease. Continued interdisciplinary collaboration will be essential to translate these insights into tangible improvements in patient care and to establish the eyelid as both a clinical marker and a therapeutic frontier in autoimmune diseases.

## Figures and Tables

**Table 1 ijms-26-11730-t001:** Immune-mediated disorders involving the eyelid skin and ocular adnexa: molecular mechanisms, clinical features, and therapeutic strategies. Abbreviations: OCP—ocular cicatricial pemphigoid; TAO—thyroid-associated orbitopathy; SLE—systemic lupus erythematosus; JAK—Janus kinase; BAFF—B-cell-activating factor; IGF-1R—insulin-like growth factor 1 receptor.

Disease	Key Molecular Mechanisms	Clinical Characteristics/Eyelid and Periocular Features	Current Therapies	Emerging/Targeted Therapies
Ocular cicatricial pemphigoid (OCP)	Autoantibodies against BP180, BP230, laminin-332; complement activation; fibrosis	Conjunctival scarring, symblepharon, entropion, trichiasis, eyelid erythema/keratinization, teleangiectasia	Systemic corticosteroids, antimetabolites, cyclophosphamide	Rituximab, JAK inhibitors, topical tacrolimus, siRNA-based therapies
Pemphigus vulgaris/Bullous pemphigoid	IgG autoantibodies against desmogleins/hemidesmosomal proteins → acantholysis and blistering	Eyelid erosions, erythematous plaques, crusting, secondary infection	Corticosteroids, immunosuppressants	Rituximab, TNF-α inhibitors, JAK inhibitors
Thyroid-associated orbitopathy (TAO)	TSHR and IGF-1R activation in fibroblasts; oxidative stress; NF-κB signaling	Eyelid retraction (hallmark), edema, dermopathy	Corticosteroids, orbital decompression surgery	Teprotumumab (IGF-1R inhibitor), tocilizumab, selenium supplementation
Systemic lupus erythematosus (SLE)	Immune complex deposition, complement activation, IFN-α pathways	Discoid lesions, erythema, scaling, dyspigmentation in periocular skin, teleangiectasia	Corticosteroids, antimalarials, immunosuppressants	Biologics targeting IFN pathways (e.g., anifrolumab)
Sjögren’s syndrome	Th17 cytokines, B-cell hyperactivity, glandular infiltration	Meibomian gland dysfunction, blepharitis, keratoconjunctivitis sicca	Topical cyclosporine, systemic immunosuppressants	JAK inhibitors, biologics (anti-BAFF)
Sarcoidosis	Th1 cytokines (TNF-α, IL-2, IFN-γ) → granuloma formation	Eyelid nodules, erythema, edema	Corticosteroids, immunosuppressants	Anti-TNF agents (infliximab, adalimumab)
Dermatomyositis	Complement-mediated microangiopathy, anti-Mi-2 autoantibodies	Violaceous periorbital (heliotrope) rash with edema	Corticosteroids, methotrexate	Biologic and JAK-targeted agents
Drug-induced eyelid dermatitis (immune-mediated hypersensitivity)	Drug-induced hypersensitivity; activation of T-cell-mediated and cytokine-driven inflammatory pathways following topical or systemic drug exposure	Erythema, edema, desquamation, and pruritus of the eyelids; may mimic chronic blepharitis; typically resolves after withdrawal of the offending agent	Drug discontinuation, topical corticosteroids, calcineurin inhibitors; avoidance of re-exposure	Identification of culprit allergens; ongoing EAACI-guided protocols for hypersensitivity management
VEXAS syndrome (autoinflammatory)	Somatic mutations in *UBA1* leading to autoinflammatory activation of myeloid lineage and cytokine overproduction (IL-6, TNF-α)	Recurrent eyelid and facial swelling, erythema, cartilage inflammation, systemic symptoms (fever, anemia, cytopenia)	Systemic corticosteroids, immunosuppressants, anti-IL-6 therapy	JAK inhibitors, targeted cytokine blockade, hematopoietic stem-cell therapy under investigation

Note: Conditions included in this table encompass autoimmune, autoinflammatory, and other immune-mediated disorders of the eyelid and ocular adnexa.

**Table 2 ijms-26-11730-t002:** Diagnostic modalities in autoimmune-mediated eyelid inflammation: clinical role, molecular insights, and translational potential. Abbreviations: AI—artificial intelligence; AS-OCT—anterior segment optical coherence tomography; BP—bullous pemphigoid; CAS—clinical activity score; DIF—direct immunofluorescence; IL—interleukin; IVCM—in vivo confocal microscopy; MMP—matrix metalloproteinase; OCP—ocular cicatricial pemphigoid; OCT—optical coherence tomography; TAO—thyroid-associated orbitopathy; TNF-α—tumor necrosis factor alpha; TSI—thyroid-stimulating immunoglobulin.

Diagnostic Modality	Clinical Role	Key Molecular/ Pathophysiological Insights	Limitations	Translational/Future Applications
Slit-lamp biomicroscopy	First-line evaluation of eyelid margin and ocular surface	Detects telangiectasia, scarring, trichiasis, keratinization; useful in grading systems (e.g., Foster, CAS)	Operator-dependent; limited for subclinical changes	Standardized digital documentation for AI analysis
Histopathology (biopsy)	Established diagnostic method for OCP, pemphigus, BP	Subepithelial fibrosis, acantholysis, granulomas	Invasive, risk of scarring	Correlation with molecular biomarkers; digital pathology
Direct immunofluorescence (DIF)	Differentiates autoimmune blistering diseases	Linear IgG, IgA, C3 deposition (OCP); intercellular IgG (pemphigus)	Requires fresh tissue; false negatives possible	Multiplex immunoassays; integration with autoantibody panels
In vivo confocal microscopy (IVCM)	Non-invasive imaging of conjunctiva and eyelid	Visualizes epithelial changes, inflammatory infiltrates, subclinical fibrosis	Limited penetration depth; requires expertise	Monitoring therapy response; AI-assisted image recognition
Anterior segment OCT (AS-OCT)	Imaging of eyelid margin and conjunctiva	Detects conjunctival scarring, eyelid margin thickening	Lower resolution than IVCM	Quantitative endpoint in clinical trials; machine learning analysis
Tear proteomics	Detects biomarkers correlating with disease activity	Elevated IL-6, IL-17, TNF-α, MMP-9 in OCP, TAO, Sjögren’s	Standardization lacking; sample variability	Biomarker-guided therapy selection; integration with omics
Serum biomarkers	Systemic disease monitoring	Anti-BP180, anti-laminin-332 (OCP); TSI (TAO)	Not always disease-specific	Genetic risk stratification; personalized treatment algorithms
Artificial intelligence (AI) tools	Automated detection, remote monitoring	Integrates imaging + clinical + biomarker data	Still experimental; requires large datasets	Prediction of flares; personalized treatment algorithms

## Data Availability

No new data were created or analyzed in this study. Data sharing is not applicable to this article.

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
