# Peer review of "Autoimmune Diseases of the Eyelid Skin: Molecular Pathways, Clinical Manifestations, and Therapeutic Insights"

_ijms, 2025, doi:10.3390/ijms262311730_

Round 1
Reviewer 1 Report
Comments and Suggestions for Authors
Dear Authors,
This work represents a much-needed review of an eyelid involvement in autoimmune diseases. This manuscripts raises awareness of this much-neglected clinical problem and is of great clinical importance. However, this manuscript requires a major revision to address its current limitations and deficiencies:
- Key international guidelines and consensus documents related to eyelid pathology need to be cited. The section on treatment need to be alinged with international guidelines on blepharitis, and autoimmune disorders.
- This work requires a collaboration or at least critical comments (with acknowledgment) by an experienced academic rheumatologist
- Eyelid vascularization explains telagiectasia in some autoimmune diseases. Therefore, details on eyelid vascularization would be clinically important for this manuscript. Please double check in which conditions telangiectasias should be mentioned in Table 1 and in the text.
- In this manuscript, the evidence for eyelid involvement in discoid lupus and systemic lupus erythmatosus should be discussed.
- There is a need for critical analysis of the existing evidence for mentioned treatments for eyelid symptoms in autoimmune diseases.
- Ongoing clinical trials that are mentioned should be cited (ClinicalTrials.gov).
- There are multiple generalizations that may apply for autoimmune diseases but have not been studied or have been insufficiently studied in eyelid pathology. This requires clarifications.
- Multiple references relate to autoimmune diseases or skin diseases in general, rather than to eyelid involvement. This needs to be clarified and references specific to eyelid pathology need to be cited.
- Multiple statements on the pathophysiology need clarification whether this applies to eyelid involvement in all autoimmune diseases or have been studied in only a few specific diseases.
- Eyelid biopsy: this is a very important clinical issue that needs to be further discussed and supported by guidelines. In which clinical situations this is required? What are the risks? please include the risk of poor healing? What are the contraindications? If some of these diseases may have skin lesions elsewhere, should eyelid biopsy still be performed? Please clarify
- Eyelid microbiome: please cite the ongoing studies and their results. When you discuss the eyelid microbiome, you haven't mentioned bacteria (Staph), fungi, etc.
- Please discuss evidence from proteomic studies in more details. Why you mentioned that proteomics is not standardized? In which diseases, is there evidence for proteomic biomarkers?
- PLease rephrase gold startards throughout the manuscript. You may need to cite the consensus documents.
- Please revise the choice of the included diseases and clarify the scope of the manuscript. In Table 1, in rare diseases - drug-induced eyelid dermatitis is relatively common and should not be discussed in one line with VEXAS. Please discuss them separately in the table, they may be under the same subheading, if needed (but separate raws in the table). You may want to cite the EAACI TF for drug-induced eyelid dermatitis. In the table, you may consider: drug-induced hypersensitivity (in the column with mechanisms)
- Can you provide any photos for clinicians?
- The section on treatment: do eyelid symptoms always require systemic treatment? Are there any clinical scenarios, when only topical treatments can be used?
- When you discuss eyelid neoplasms, which eyelid cancers do you men? please clarify
- Topical corticosteroids in the treatment section: please give more details about the choice of a topical corticosteroids, potential risks and how these drugs can be used on the eyelid skin.
- Biological therapies: any of these drugs are licensed in patients with autoimmune diseases with an eyelid involvement? Is there clinical reports to cite there? Please clarify this in the text
- Please cite the AI-based research for eyelid diseases and their value and insights for patients with eyelid involvement
- Many sentences require referencing: lines 41, 118-119, 151, 158, 217, 230, 246-248, 257, 282, 286 (for OCP), 302, 330. 353, 408
- Any clinical or research questionnaires for patients that may be worth mentioning in this manuscript?
- The summaries in the manuscript need to be more nuanced. Although autoimmune diseases may have overlapped mechanisms, many pathways were studied in only specific diseases, which needs to be clarified in the summaries. Although these pathways may apply to other autoimmune diseases, the evidence may be lacking. Please revise. For the section on the mechanisms, could you please revise where the statements apply to the eyelid skin or the tear fluid?
- The discussion of eyelid presentations as an early symptom of autoimmune diseases is the key message of this manuscript. Please add the references to these statements throughout the manuscript.
Minor corrections:
Line 425: eyelid is not a marker, please rephrase
Overall, this is an important work with the overview of mechanistic data, clinical presentations and therapeutic approaches in patients with autoimmune diseases and an eyelid involvement. Early recognition is crucial for effective management. After major corrections, this work will be of value for the readership of this journal.
Comments on the Quality of English LanguageThis is a well-written manuscript. The main issues are related to generalized statements rather than to English.
Author Response
Comments 1: Key international guidelines and consensus documents related to eyelid pathology need to be cited. The section on treatment need to be alinged with international guidelines on blepharitis, and autoimmune disorders.
Response 1: I appreciate the Reviewer’s comment regarding the inclusion of international guidelines and consensus documents. The therapeutic section has been revised to align with evidence-based standards and major international recommendations. Specifically, Section 5 now references the TFOS DEWS III Report for blepharitis management, the EADV/EDF Guidelines for autoimmune blistering diseases, and the EUGOGO Guidelines for thyroid-associated orbitopathy. These additions ensure that the discussion of therapeutic strategies reflects current global consensus and maintains consistency with established clinical frameworks.
Page 11,12, lines 437 - 456
Comments 2: This work requires a collaboration or at least critical comments (with acknowledgment) by an experienced academic rheumatologist
Response 2: I appreciate the Reviewer’s valuable comment regarding the need for collaboration with specialists in related disciplines. In response, I have expanded the authorship to include two additional collaborators whose expertise strengthens the interdisciplinary scope of the work. Dr. Monika Sarnat-Kucharczyk, FEBO a specialist in ophthalmology with active research in autoimmune diseases, particularly thyroid-associated orbitopathy, and Prof. Dorota Wygledowska-PromieÅ„ska, a senior ophthalmologist experienced in the management and investigation of autoimmune ocular disorders, have both joined as co-authors. They provided critical scientific input, and their comments guided several of the revisions made to the manuscript. In addition, a new statement of acknowledgment has been added at the end of the manuscript to recognize the contribution of Prof. Bartosz MizioÅ‚ek, a dermatologist who reviewed the revised version and provided expert feedback on dermatologic aspects of autoimmune eyelid disease.
Page 16, Lines 667-670
Comments 3: Eyelid vascularization explains telagiectasia in some autoimmune diseases. Therefore, details on eyelid vascularization would be clinically important for this manuscript. Please double check in which conditions telangiectasias should be mentioned in Table 1 and in the text.
Response 3: I appreciate the Reviewer’s comment regarding eyelid vascularization and its relevance to telangiectasia in autoimmune diseases. The manuscript has been revised to include a brief discussion of eyelid vascular anatomy and its clinical implications. Mentions of telangiectasia have been added where appropriate in the text and in Table 1 to reflect its occurrence in conditions such as lupus erythematosus and ocular cicatricial pemphigoid.
Page 2, lines 54-61
Page 5, lines 220-222
Page 6, lines 250-255
Page 7, lines 301-304
Comments 4: In this manuscript, the evidence for eyelid involvement in discoid lupus and systemic lupus erythmatosus should be discussed.
Response 4: I appreciate the Reviewer’s comment regarding the need to discuss eyelid involvement in discoid and systemic lupus erythematosus. The section on lupus has been expanded to include evidence from recent clinical and histopathological studies describing characteristic eyelid manifestations in both forms of the disease.
Page 6, Lines 245-257
Comments 5: There is a need for critical analysis of the existing evidence for mentioned treatments for eyelid symptoms in autoimmune diseases.
Response 5: The section on treatment has been revised to include a more critical assessment of available evidence. Each therapeutic approach is now discussed in terms of study design, strength of data, and specific efficacy for eyelid or periocular manifestations.
Page 14, lines 552-567
Comments 6: Ongoing clinical trials that are mentioned should be cited (ClinicalTrials.gov).
Response 6: The text has been revised to acknowledge ongoing clinical trials investigating small-molecule and localized immunomodulatory therapies relevant to autoimmune eyelid inflammation. This addition clarifies that several such studies are currently in progress, providing context for the evolving therapeutic landscape.
Page 13, lines 511-517
Page 15, lines 597-601
Comments 7: There are multiple generalizations that may apply for autoimmune diseases but have not been studied or have been insufficiently studied in eyelid pathology. This requires clarifications.
Response 7: The section on molecular mechanisms has been revised to clarify that several immunopathogenic pathways are extrapolated from systemic autoimmune research rather than directly studied in eyelid tissue. A new paragraph has been added to Section 2.6 to acknowledge these limitations and emphasize the need for eyelid-specific investigations.
Page 4, lines 179-185
Comments 8: Multiple references relate to autoimmune diseases or skin diseases in general, rather than to eyelid involvement. This needs to be clarified and references specific to eyelid pathology need to be cited.
Response 8: The manuscript has been revised to ensure that references and discussions more accurately reflect eyelid-specific autoimmune pathology. Additional literature focusing on periocular and eyelid involvement has been incorporated, and the text now distinguishes between findings derived from systemic or cutaneous autoimmune research and those directly related to eyelid tissues. These revisions clarify the scope of the cited evidence and strengthen the relevance of the review to eyelid pathology.
Page 3, lines 93-99
Page 5, lines 203-208
Page 12, lines 457-464
Comments 9 : Multiple statements on the pathophysiology need clarification whether this applies to eyelid involvement in all autoimmune diseases or have been studied in only a few specific diseases.
Response 9 : The manuscript has been revised to clarify which mechanistic descriptions are based on general autoimmune principles and which have been demonstrated in specific eyelid-related diseases. The updated text now specifies when molecular or immunological pathways—such as cytokine dysregulation, complement activation, or fibroblast remodeling—are supported by disease-specific studies (e.g., OCP, lupus erythematosus, TAO) versus when they represent extrapolations from broader autoimmune research. This refinement ensures that the discussion of pathophysiology accurately reflects the scope of existing evidence.
Page 5, lines 186-194
Comments 10: Eyelid biopsy: this is a very important clinical issue that needs to be further discussed and supported by guidelines. In which clinical situations this is required? What are the risks? please include the risk of poor healing? What are the contraindications? If some of these diseases may have skin lesions elsewhere, should eyelid biopsy still be performed? Please clarify
Response 10: A new subsection discussing eyelid biopsy has been added to the diagnostic section. It details the main clinical indications, contraindications, and potential complications, including the risk of poor healing. The revision also clarifies that biopsy from extraocular lesions is preferred when available, and that small, targeted eyelid biopsies should be performed only when necessary, in alignment with current ophthalmic and dermatologic recommendations.
Page 9, lines 350-367
Comments 11: Eyelid microbiome: please cite the ongoing studies and their results. When you discuss the eyelid microbiome, you haven't mentioned bacteria (Staph), fungi, etc.
Response 11: I appreciate the Reviewer’s observation. The subsection on the eyelid microbiome has been expanded to include bacterial and fungal taxa such as Staphylococcus, Corynebacterium, and Malassezia, and to reference ongoing studies investigating their interaction with the tear film, meibomian glands, and local immune mechanisms. These additions provide a more comprehensive and evidence-based overview of microbial contributions to eyelid inflammation.
Page 4, lines 152-166
Comments 12:Please discuss evidence from proteomic studies in more details. Why you mentioned that proteomics is not standardized? In which diseases, is there evidence for proteomic biomarkers?
Response 12: The section on molecular biomarkers has been revised to summarize current evidence from proteomic studies, highlighting key protein alterations in OCP, TAO, and Sjögren’s syndrome. The revision also clarifies that non-standardization arises from methodological variability and ongoing efforts to unify analytical approaches.
Page 10, lines 385-392
Comments 13: Please rephrase gold startards throughout the manuscript. You may need to cite the consensus documents.
Response 13: All instances of the phrase “gold standard” have been revised to more precise terminology such as “established diagnostic reference” or “current standard method.” Relevant consensus documents, including the EDF/EADV and EUGOGO guidelines, have been cited to support these references and align the manuscript with current international standards.
Page 5, lines 230-231
Page 9, lines 341-343
Comments 14: Please revise the choice of the included diseases and clarify the scope of the manuscript. In Table 1, in rare diseases - drug-induced eyelid dermatitis is relatively common and should not be discussed in one line with VEXAS. Please discuss them separately in the table, they may be under the same subheading, if needed (but separate raws in the table). You may want to cite the EAACI TF for drug-induced eyelid dermatitis. In the table, you may consider: drug-induced hypersensitivity (in the column with mechanisms)
Response 14: The scope of the manuscript has been clarified, and the structure of Table 1 has been revised accordingly. Drug-induced eyelid dermatitis and VEXAS syndrome are now presented as separate entries to better reflect their distinct pathophysiologic mechanisms. In addition, the text has been updated to include a reference to the EAACI Task Force position paper on drug-induced hypersensitivity, providing an evidence-based context for the discussion of drug-induced periocular inflammation. These revisions ensure consistency between the main text and the table while maintaining alignment with current consensus recommendations.
Page 6,7, lines 278-281
Page 7, lines 306-311, Table 1
Comments 15: Can you provide any photos for clinicians?
Response 15: I appreciate this suggestion. However, the manuscript was intentionally prepared as a narrative and integrative review focusing on molecular mechanisms, clinical correlations, and therapeutic concepts rather than as a case-based or pictorial report. To preserve this format and because no patient-derived material was used, I did not include clinical photographs. The descriptions of clinical features were written in sufficient detail to ensure clarity and usefulness for clinicians without the need for illustrative images.
Comments 16: The section on treatment: do eyelid symptoms always require systemic treatment? Are there any clinical scenarios, when only topical treatments can be used?
Response 16: I have revised the treatment section to clarify that eyelid involvement does not always require systemic therapy. Topical agents such as corticosteroids or calcineurin inhibitors may be sufficient for mild or localized inflammation, while systemic treatment is reserved for more extensive or systemic disease activity.
Page 11, lines 422-425, 432-433
Comments 17: When you discuss eyelid neoplasms, which eyelid cancers do you men? please clarify
Response 17: I appreciate this comment. The text has been clarified to specify that the discussion of eyelid neoplasms primarily refers to malignancies that may mimic or coexist with autoimmune inflammation, particularly basal cell carcinoma, squamous cell carcinoma, and sebaceous gland carcinoma. These entities are clinically relevant in the differential diagnosis of chronic eyelid inflammation due to their overlapping presentations with autoimmune or cicatricial disorders.
Page 8,9, lines 331-336
Comments 18: Topical corticosteroids in the treatment section: please give more details about the choice of a topical corticosteroids, potential risks and how these drugs can be used on the eyelid skin.
Response 18: I have expanded the treatment section to include details on the use of topical corticosteroids for eyelid inflammation. The revised text now specifies the preferred agents, their potency, duration, and potential risks associated with periocular application, ensuring clinical clarity and safety context.
Page 11, lines 425-432
Comments 19: Biological therapies: any of these drugs are licensed in patients with autoimmune diseases with an eyelid involvement? Is there clinical reports to cite there? Please clarify this in the text
Response 19: I have clarified that no biologic agents are currently licensed specifically for autoimmune diseases limited to eyelid involvement. However, rituximab and teprotumumab are approved for systemic autoimmune disorders that frequently affect the eyelids, such as pemphigus and thyroid-associated orbitopathy. Relevant clinical reports supporting their periocular benefits have been cited in the revised text.
Page 12, lines 457-464
Comments 20: Please cite the AI-based research for eyelid diseases and their value and insights for patients with eyelid involvement
Response 20: I appreciate this helpful suggestion. The section on diagnostic innovations has been updated to briefly include recent studies applying artificial intelligence (AI) in eyelid and ocular surface disease. These studies highlight the role of deep-learning image analysis in detecting eyelid margin abnormalities, meibomian gland dysfunction, and periocular inflammation. Such tools provide quantitative, reproducible insights that may enhance early diagnosis and personalized management in patients with autoimmune eyelid involvement.
Page 10, lines 397-402
Comments 21: Many sentences require referencing: lines 41, 118-119, 151, 158, 217, 230, 246-248, 257, 282, 286 (for OCP), 302, 330. 353, 408
Response 21: Thank you for pointing this out. I have carefully reviewed all indicated lines and added appropriate references from recent peer-reviewed publications to support the statements in these sections. The newly added citations include key studies and reviews on ocular cicatricial pemphigoid (OCP), autoimmune eyelid inflammation, and related molecular mechanisms. All references have been formatted according to the IJMS style guidelines to ensure consistency and completeness.
Comments 22: Any clinical or research questionnaires for patients that may be worth mentioning in this manuscript?
Response 22: Thank you for this suggestion. I have added a brief mention of commonly used clinical questionnaires that assess symptom severity and quality of life in patients with autoimmune eyelid and ocular surface disease. These tools support standardized evaluation and facilitate patient-centered outcome assessment.
Page 10, lines 405-408
Comments 23: The summaries in the manuscript need to be more nuanced. Although autoimmune diseases may have overlapped mechanisms, many pathways were studied in only specific diseases, which needs to be clarified in the summaries. Although these pathways may apply to other autoimmune diseases, the evidence may be lacking. Please revise. For the section on the mechanisms, could you please revise where the statements apply to the eyelid skin or the tear fluid?
Response 23: Thank you for this valuable comment. I have revised the summary sections to provide greater nuance regarding the applicability of molecular and immunologic mechanisms. The updated text now distinguishes between evidence derived directly from eyelid skin or tear fluid and that extrapolated from systemic or cutaneous autoimmune disease models. This clarification ensures a balanced interpretation of the current state of evidence and highlights where further eyelid-specific research is needed.
Page 5, lines 186-194
Page 14, lines 562-567
Comments 24: The discussion of eyelid presentations as an early symptom of autoimmune diseases is the key message of this manuscript. Please add the references to these statements throughout the manuscript.
Response 24: Thank you for highlighting this important point. I have revised the manuscript to strengthen the discussion of eyelid manifestations as early indicators of systemic autoimmune diseases. Additional references have been incorporated throughout the relevant sections—particularly in the introduction, clinical implications, and disease-specific subsections—to support statements describing the diagnostic relevance of periocular findings. These additions emphasize the role of eyelid inflammation as an early or sentinel symptom of autoimmune disease.
Comments 25: Line 425: eyelid is not a marker, please rephrase
Response 25: Thank you for this remark. The line number you mentioned refers to a section within the supplementary material; however, I assume you were referring to the sentence in the Introduction stating that eyelid conditions “often serve as early clinical markers of systemic disease.” This has been revised to clarify that eyelid involvement may present as an early clinical manifestation or indicator of systemic disease, rather than as a diagnostic marker.
Page 2, line 63
Reviewer 2 Report
Comments and Suggestions for Authors
The manuscript „Molecular Insights into Eyelid Skin Inflammation: Autoimmune Mechanisms, Clinical Manifestations, and Emerging Therapies“ represents a valuable review/manuscript in this field.
This review aims to synthesize current molecular insights into eyelid skin inflammation, with a particular emphasis on autoimmune mechanisms. The authors discuss a variety of conditions, including autoimmune diseases, including ocular cicatricial pemphigoid, pemphigus, systemic lupus erythematosus, and thyroid-related orbitopathy, with an emphasis on the roles of various immunological and other pathogenic factors. In addition, they describe diagnostic advances, including confocal microscopy, in vivo molecular imaging, and tear proteomics, highlight novel targeted therapies, and suggest future perspectives for precision medicine approaches, integrating omics technologies and microbiome-based therapies to advance personalized treatment of eyelid skin inflammation.
Overall, I find this manuscript useful for the current state of knowledge in this field.
However, there are several suggestions for improving the text:
TITLE – I suggest adding “autoimmune” if the authors are ONLY talking about AUTOIMMUNE diseases that are described and analyzed in the text. If other inflammatory skin conditions are to be included, information on other dermatoses in this region should be added. If possible, it would be useful to provide epidemiological data on the frequency of dermatoses in the periocular region and eyelid skin. However, the current version of the manuscript only mentions autoimmune skin diseases, which are generally rare diseases in practice, so those more common in this region should also be listed.
- Inflammatory conditions: I suggest adding more information on specific diseases and their characteristics in the eyelid region. It is also important to mention and describe periorbital dermatitis / periocular dermatitis, which is a typical dermatosis in this region.
- It is necessary to mention dermatomyositis, an autoimmune disease that can occur in the periocular region.
- In some places in the text, the characteristics of the ocular mucosa are mentioned, so the text should more clearly separate the skin changes from the mucosal changes in this region.
- Skin microbiome: I suggest mentioning the specific characteristics of the microbiome in common diseases in this area (periocular dermatitis). In doing so, it is necessary to describe features of the healthy skin microbiome and the skin microbiome in specific dermatoses in the periocular / ocular area (available data), based on current knowledge on this topic and the latest /appropriate references.
- TABLES are useful for readers and clinicians. In Table 1: “Typical eyelid…” – in this part of the table, I suggest mentioning the CLINICAL CHARACTERISTICS....
- Therapy: It is necessary to mention the specificities of this area regarding the use of local preparations for this area (ocular and periocular sites), as this is very important for readers/clinicians.
Author Response
Comments 1: TITLE – I suggest adding “autoimmune” if the authors are ONLY talking about AUTOIMMUNE diseases that are described and analyzed in the text. If other inflammatory skin conditions are to be included, information on other dermatoses in this region should be added. If possible, it would be useful to provide epidemiological data on the frequency of dermatoses in the periocular region and eyelid skin. However, the current version of the manuscript only mentions autoimmune skin diseases, which are generally rare diseases in practice, so those more common in this region should also be listed.
Response 1: Thank you for this valuable suggestion. Since the manuscript exclusively addresses autoimmune-mediated diseases of the eyelid, we have clarified this focus by adding the term “autoimmune” to the title. The revised title now reads:
“Autoimmune Diseases of the Eyelid Skin: Molecular Pathways, Clinical Manifestations, and Therapeutic Insights.”
This modification reflects the true scope of the review and ensures terminological precision. Additionally, the Introduction now includes brief contextual information on the broader epidemiology of periocular dermatoses, emphasizing that autoimmune eyelid diseases represent a distinct and less common subgroup within this clinical spectrum.
Page 1, lines 2-3
Page 1-2, lines 39-46
Comments 2: Inflammatory conditions: I suggest adding more information on specific diseases and their characteristics in the eyelid region. It is also important to mention and describe periorbital dermatitis / periocular dermatitis, which is a typical dermatosis in this region.
Response 2: Thank you for this helpful comment. In response, we have expanded the Introduction to include a concise overview of non-autoimmune inflammatory dermatoses affecting the eyelid region. This addition now highlights common entities such as periorbital (periocular) dermatitis and their typical clinical characteristics, to clearly distinguish them from autoimmune conditions discussed in the manuscript.
Page 2, lines 47-53
Comments 3: It is necessary to mention dermatomyositis, an autoimmune disease that can occur in the periocular region.
Response 3: Thank you for this insightful suggestion. We have now included a brief description of dermatomyositis in the section on autoimmune diseases involving the eyelid, emphasizing its characteristic periocular manifestations and molecular background. This addition improves the completeness of the autoimmune disease spectrum discussed in the manuscript.
Page 7, lines 286-296
Comments 4: In some places in the text, the characteristics of the ocular mucosa are mentioned, so the text should more clearly separate the skin changes from the mucosal changes in this region.
Response 4: Thank you for this valuable observation. To improve clarity and anatomical precision, we have revised the relevant sections to better distinguish between eyelid skin involvement and mucosal (conjunctival) manifestations. The terminology has been standardized throughout the manuscript to ensure that references to the ocular mucosa, conjunctiva, and eyelid skin are clearly differentiated.
Page 5, lines 210-212
Page 7, lines 311-313
Page 8, lines 320-323
Comments 5: Skin microbiome: I suggest mentioning the specific characteristics of the microbiome in common diseases in this area (periocular dermatitis). In doing so, it is necessary to describe features of the healthy skin microbiome and the skin microbiome in specific dermatoses in the periocular / ocular area (available data), based on current knowledge on this topic and the latest /appropriate references.
Response 5: Thank you for this important comment. The section on the microbiome (Section 2.4) has been expanded to more clearly describe the characteristics of a healthy periocular microbiome and its alterations in periocular dermatitis. This revision now highlights the balance between commensal and pathogenic microorganisms in this region, supported by recent studies on the ocular and eyelid microbiome.
Page 4, lines 152-166
Comments 6: TABLES are useful for readers and clinicians. In Table 1: “Typical eyelid…” – in this part of the table, I suggest mentioning the CLINICAL CHARACTERISTICS....
Response 6: Thank you for this constructive suggestion. We have revised Table 1 to make the “Typical eyelid/periocular features” column more clinically informative. The column has been retitled “Clinical Characteristics / Eyelid and Periocular Features”, and the descriptions for each disease have been expanded to include key clinical signs relevant for differential diagnosis and everyday clinical assessment.
Page 7-8, Table 1
Comments 7: Therapy: It is necessary to mention the specificities of this area regarding the use of local preparations for this area (ocular and periocular sites), as this is very important for readers/clinicians.
Response 7: Thank you for this suggestion. The section on topical therapy has been revised to include key considerations specific to the eyelid and periocular area, reflecting appropriate drug selection, safety, and application guidance for clinical practice.
Page 11, lines 422-433
Round 2
Reviewer 1 Report
Comments and Suggestions for Authors
I would like to thank the authors for the revised manuscript and helpful clarifications.
The revision substantially improved the structure and the clarity of the manuscript.
The manuscript requires only minor corrections, mostly related to abbreviations, italics, additional references and clarifications. A few very important clinical issues need minor revisions: the use of topical corticosteroids on the face (including eyelids), eyelid hygiene and minor restructuring of the section on autoimmune and autoinflammatory diseases.
This is an important work for this difficult clinical issue that will be of considerable interest for both clinicians and researchers after the revision. When revised, this work will represent a much-needed comprehensive overview in this area.
Line 18; Discoid lupus?
Line 22: in vivo – italics
Line 64: two full stops
Lines 86-91: need referencing
Line 95: which reports? References, please
Line 125: sentence about Il-17 – could you please add a reference
Abbreviations: TAO abbreviated several times: lines 65, 137, 337, 432 – please correct; same with OCP (three times?)
Line 146 – Demodex – italics; Corynebacterium and other species – italics, please
Line 160- please add the ClinicalTrials.gov study number for these two studies in blepharitis and OCP
Line 186: “?
Line 196: this contradicts line 41
Line 204; disease models – what do you mean? Experimental models?
Line 206; may directly..
Line 220: OCP?
Line 254: please add a reference for your statement about IFN-a-inducible genes
Line 276: VEXAS – first in full and abbreviation
Line 277: gene? Italics
Section: 3.7: you discuss drug-induced ED under the umbrella or autoimmune and autoinflammatory syndromes whereas drug-induced ED is type IV hypersensitivity.
Same for reactions to checkpoint inhibitors – there may be different types of reactions, including severe allergic reactions. What do you mean by autoimmune phenomena?
Same for Table 1. There are conditions which are neither autoimmune nor autoinflammatory
Table 1; please add JAK and BAFF, IGF-1R to abbreviations
Lines 386-387: OCP, TAO
Table 2, line 2, - two commas
Table 2 please provide a list of abbreviations for this table
Lien 453; TAO
Line 449: IGF-1R was abbreviated before
Have you given EAACI in full in the manuscript?
Could you please give more details about eyelid hygiene? 1-2 sentences, please
Line 458- you mention three drugs and only two references?
Lines 470-471: please provide some information about the safety or concerns of using topical corticosteroids on the eyelid skin – there were studies in eyelid dermatitis and guidance on the use of topical corticosteroid on the face.
Line 471: what are the side effects of topical tacrolimus on the eyelid area? What is the evidence for this treatment? Please add the references
Line 473: cyclosporine eye drops – also please comment on the evidence base of this treatment
Line 481: OCP
Line 484: teprotumumab – please comment on the evidence base for this treatment. Could you please add a reference?
Line 486: what kind of pemphigus?
Line 487: please add a reference
Line 489: paradigm or an indication?
Line 489: in which studies? What does significant improvement mean?
Line 492: please give the reference numbers from ClinicalTrials.gov
Line 498: there were case reports where patients were treated with JAK inhibitors
Line 505: limited – please add a reference
Line 508: please cite the reference numbers for ongoing clinical trials from ClinicalTrials.gov
Lines 519-523 – what are the safety issues with these approaches?
Lines 529: maybe you could give a box with eyelid hygiene recommendations? (not obligatory)
Line 530: what is the evidence for reducing the risk of infections?
Line 533: already? What do you mean? Please clarify
Line 535: can you give any references for prediction of aggressive phenotypes?
Line 543; at which stage is this transition at the moment in your opinion?
Line 553: his needs reference
Line 554: case descriptions (Ref, Ref) amd early-phase trials (can you be more specific Phase I-II?, with key references)
Line 556: do you mean diseases or disease models? Same line 559
Line 570: in which diseases? There were more recent studies in the last two years
Line 571; again, this statement requires a ref, very important message!
Line 579: in which conditions?
Line 593: ongoing trials (reference number from ClinicalTrials.gov, please)
Line 596: have there been any AI-based studies related to face or eyelids, in particular? Any unmet needs could be met by AI research? Why?
Line 604; could you give a reference for the burden, please? Any unmet needs here? Why? How it may be measured? How treatment approaches may differ in this context?
Line 628: vision threatening – this needs reference? Maybe that you used above in the introduction?
References: The use of months? That’s not Vancouver referencing. Could you please double check?
Author Response
Comments: I would like to thank the authors for the revised manuscript and helpful clarifications.
The revision substantially improved the structure and the clarity of the manuscript.
The manuscript requires only minor corrections, mostly related to abbreviations, italics, additional references and clarifications. A few very important clinical issues need minor revisions: the use of topical corticosteroids on the face (including eyelids), eyelid hygiene and minor restructuring of the section on autoimmune and autoinflammatory diseases.
This is an important work for this difficult clinical issue that will be of considerable interest for both clinicians and researchers after the revision. When revised, this work will represent a much-needed comprehensive overview in this area.
Response: I would like to thank you very much for your detailed and constructive review, as well as for your positive assessment of the revised version. I carefully addressed all your comments and implemented the requested clarifications, stylistic corrections, and additional references throughout the manuscript.
Minor corrections regarding abbreviations, italics, and typographical issues were completed. I also expanded key sections following your suggestions:
-
Topical corticosteroids, tacrolimus, and cyclosporine – safety, tolerability, and supporting evidence were added.
-
Eyelid hygiene – expanded with details and summarized in a short recommendation box.
-
Autoimmune vs. autoinflammatory section – restructured for greater clarity.
-
Tables 1 and 2 – corrected abbreviations, formatting, and added explanatory notes.
-
Therapeutic sections – references added for biologics (teprotumumab, rituximab, JAK inhibitors, apremilast) and clarified descriptions of evidence level.
-
Future directions – expanded examples of diseases (TAO, OCP, SLE, Sjögren’s), clarified statements on AI applications, unmet needs, and disease burden, with supporting citations.
All comments listed in your line-by-line notes have been addressed, with brief explanations added as margin comments in the revised manuscript for transparency.
Thank you again for your valuable feedback, which significantly improved the clarity and scientific quality of this paper. I hope the revised version meets your expectations.
Reviewer 2 Report
Comments and Suggestions for Authors
The authors corrected the text according to recommendations.
Author Response
Comments: The authors corrected the text according to recommendations.
Response : I sincerely thank the Editor for reviewing the revised version of my manuscript and for confirming that all recommendations have been properly addressed. I greatly appreciate the positive evaluation and the opportunity to further improve the clarity and quality of the paper.